# Role of Brain Derived Extracellular Vesicles in Decoding Sex Differences Associated with Nicotine Self-Administration

**DOI:** 10.3390/cells9081883

**Published:** 2020-08-11

**Authors:** Sneh Koul, Victoria L. Schaal, Subhash Chand, Steven T. Pittenger, Neetha Nanoth Vellichirammal, Vikas Kumar, Chittibabu Guda, Rick A. Bevins, Sowmya V. Yelamanchili, Gurudutt Pendyala

**Affiliations:** 1Department of Anesthesiology, University of Nebraska Medical Center, Omaha, NE 68198, USA; sneh.koul@unmc.edu (S.K.); vicki.schaal@unmc.edu (V.L.S.); subhash.chand@unmc.edu (S.C.); syelamanchili@unmc.edu (S.V.Y.); 2Department of Psychology, University of Nebraska-Lincoln, Lincoln, NE 68588, USA; steven.pittenger@nih.gov (S.T.P.); rbevins1@unl.edu (R.A.B.); 3Department of Genetics Cell Biology and Anatomy, University of Nebraska Medical Center, Omaha, NE 68198, USA; n.nanothvellichiram@unmc.edu (N.N.V.); babu.guda@unmc.edu (C.G.); 4Mass Spectrometry and Proteomics Core Facility, University of Nebraska Medical Center, Omaha, NE 68198, USA; vikas.kumar@unmc.edu

**Keywords:** nicotine, extracellular vesicles, sex differences, proteomics, bioinformatics

## Abstract

Smoking remains a significant health and economic concern in the United States. Furthermore, the emerging pattern of nicotine intake between sexes further adds a layer of complexity. Nicotine is a potent psychostimulant with a high addiction liability that can significantly alter brain function. However, the neurobiological mechanisms underlying nicotine’s impact on brain function and behavior remain unclear. Elucidation of these mechanisms is of high clinical importance and may lead to improved therapeutics for smoking cessation. To fill in this critical knowledge gap, our current study focused on identifying sex-specific brain-derived extracellular vesicles (BDEV) signatures in male and female rats post nicotine self-administration. Extracellular vesicles (EVs) are comprised of phospholipid nanovesicles such as apoptotic bodies, microvesicles (MVs), and exosomes based on their origin or size. EVs are garnering significant attention as molecules involved in cell–cell communication and thus regulating the pathophysiology of several diseases. Interestingly, females post nicotine self-administration, showed larger BDEV sizes, along with impaired EV biogenesis compared to males. Next, using quantitative mass spectrometry-based proteomics, we identified BDEV signatures, including distinct molecular pathways, impacted between males and females. In summary, this study has identified sex-specific changes in BDEV biogenesis, protein cargo signatures, and molecular pathways associated with long-term nicotine self-administration.

## 1. Introduction

Smoking remains a significant health and economic burden in the United States, resulting in 480,000 deaths and costing more than $300 billion annually [1]. While many smokers report a desire to quit, many are unsuccessful in attempting to quit. The robustness of this addiction can be partially attributed to nicotine, the primary addictive component in cigarettes [2]. Nicotine is a well-known psychostimulant that diffuses readily into brain tissue, where it binds to nicotinic acetylcholine receptors (nAChRs) [3]. Activation of these receptors produces a wide variety of short and long-term effects on various organ systems. There are now a variety of well-known sex differences associated with smoking. Women are more likely to smoke fewer cigarettes per day compared to men, and they also have a higher nicotine dependence [4,5]. This may be due to motivations associated with smoking [6]. Generally, women also smoke in response to non-nicotine stimuli. This response to non-nicotine stimuli may help explain why women have greater difficulty in quitting compared to men. Research has also shown that men tend to improve cessation rates through nicotine interventions, while women do better with non-nicotine interventions such as counseling [7,8]. These differences suggest that women have a higher sensitivity to non-pharmacokinetic effects of nicotine [9]. Though there are many environmental and societal factors that play a role in nicotine sex differences, there are also biological factors to consider. Previous research emphasizes the developmental impact of nicotine on sexual differentiation and explains how prenatal and adolescent nicotine exposure can have sex-specific effects due to changes in circulating gonadal hormones [10]. In addition to the changes in circulating gonadal hormones, there is an impact at the molecular level with the differential expression of X and Y genes. These biological factors, as a result, impact neuronal composition and functions like memory, cognitive function, and reward processing.

In tandem, pre-clinical animal models have shown nicotine associated sex differences. Female rats self-administer nicotine at lower doses and have higher intake in socially-acquired self-administration with more motivation to obtain nicotine [11,12,13,14,15]. Female rats also show greater nicotine-induced locomotor sensitization, higher sensitivity to nicotine conditioning, and increased withdrawal signs [16]. Previous research on reinforcing effects using the nicotine self-administration task has also helped explain the associated pathways and neurotransmitter systems critical to nicotine addiction. The mesocorticolimbic pathway has been shown to play an essential role in nicotine self-administration, where the ventral tegmental area (VTA) is the primary site responsible for the reinforcing effects of nicotine [17]. Another brain region of particular interest is the prefrontal cortex (PFC), as it may play a role in sex differences associated with nicotine self-administration. For example, neurotensin, a neuropeptide, is increased in the PFC of male rats but not in females after nicotine self-administration in comparison to saline controls [9].

While research has established the likely importance of sex differences in the behavioral and neural effects of nicotine, there is a significant knowledge gap in discerning the molecular mechanisms. One such critical regulator is an extracellular vesicle (EV); extracellular vesicles are phospholipid nanovesicles that include, based on their origin or size, apoptotic bodies, microvesicles, and exosomes. These EVs are garnering significant attention as molecules involved in cell–cell communication between cells and regulating the pathophysiology of several diseases [18,19,20]. EVs carry molecular cargo such as proteins, lipids, and RNA [21,22,23]. The molecular composition of the EV-cargo generally depends on their cells/tissue of origin. Furthermore, recent works have depicted their role as next-generation biomarkers based on their feasibility as therapeutic drug-delivery nanocarriers. In immunotherapy, they have generated much excitement as candidates for diagnostics and therapeutics in an array of diseases [24,25,26,27,28]. In this current study, we performed ultrastructural characterization of BDEV, their biogenesis, and identification of signatures using quantitative mass spectrometry-based proteomics from male and female Sprague Dawley rats following either long-term nicotine or saline self-administration. Further, employing bioinformatics analysis, we identified distinct biological processes associated with differentially expressed proteins of BDEV between males and females. To our knowledge, this is the first study to identify sex-specific changes in BDEV biogenesis, protein cargo signatures, and molecular pathways associated with long-term nicotine self-administration.

## 2. Materials and Methods

### 2.1. Self-Administration and Brain Tissue Collection

The brain tissue used in this study was from a previously published work from our group [29]. Briefly, 9-week-old male and female Sprague Dawley rats were trained for lever presses following which rats were surgically implanted with a jugular catheter (IUCAC #1477 and Animal Welfare Assurance Number A3459-01). After recovery, rats received 1 h training sessions with sucrose available on a variable ratio (VR) 3 schedule of reinforcement for 1 week following which rats were randomly distributed to perform self-administration of either nicotine (0.03 mg/kg/infusion) or saline on a VR3 schedule. The self-administration sessions were conducted for 22 days. Twenty-four hours after the last self-administration session, brains were extracted and stored at −80 °C until further use. Figure 1 shows further steps carried out after brain tissue isolation.

### 2.2. BDEV Isolation

The goal here was to identify BDEVs as potential markers associated with nicotine dependency between male and female rats. Of note, we understand that the methodology to isolate extracellular vesicles should be complemented with other techniques, but there are various published studies on those fronts. In this current study, we performed BDEV isolation from rats in all four treatment groups (*n* = 4–6 per group) using the sucrose density, which is well established in our lab, as evidenced by our previous publications [30,31,32]. Briefly, brain tissue was minced and digested in 20 units/mL papain in Hibernate A (Life Technologies, Waltham, CA, USA). Digestion was then followed by the immediate addition of cold Hibernate A to stop the enzymatic digestion. Cells, large particles, and debris were removed via multiple centrifugations (300× *g*, 2000× *g*, 10,000× *g*) and filtered through a 0.22 μm filter. To ensure maximum concentration of EVs, samples were then ultra-centrifuged at 100,000× *g* for an hour at 4 °C, followed by its purification using a density gradient separation technique. Sucrose concentrations ranging from 0.25 to 2 M were used for the density gradient separation, and the concentrated EVs pellet was resuspended in the 0.95 M layer and ultra-centrifuged at 200,000× *g* for 16 h at 4 °C. Fractions enhanced with EVs were subjected to additional ultracentrifugation at 100,000× *g* to obtain the purified EVs pellet. These EV pellets were then re-suspended in 1× particle-free phosphate-buffered saline (PBS) and protein content measured using bicinchoninic acid assay (BCA).

### 2.3. Western Blot: BDEV Marker Validation

The purity of BDEVs was determined through specific antibodies described in previous studies [30,31,32]. Briefly, BDEV protein lysates were prepared using RIPA buffer with 1% SDS and protease-phosphatase inhibitor following which 40 μg of protein (for positive markers) and 10 μg (for the negative marker) from a control brain run on 4% to 12% Bis-Tris gels (Invitrogen, Waltham, MA, USA) under reducing (Hsp70, flotillin, and calnexin) and non-reducing conditions (CD81) followed by transfer using iBLOT2 (Invitrogen). Nonspecific antibody binding was done using 5% nonfat dried milk. Immunoblotting was carried out with primary antibodies at 4 °C against the positive BDE markers Hsp70 (1:1000, Sigma, St. Louis, MO, USA), flotillin-1 (1:1000, Abcam, Cambridge, UK), CD81 (1:500, Bio-Rad, Hercules, CA, USA), and the negative marker calnexin (1:1000, Abcam) followed by secondary antibody (1:2500 HRP conjugated anti-mouse IgG for Hsp70, 1:2500 HRP conjugated anti-rabbit IgG for flotillin-1, 1:1500 HRP conjugated anti-hamster IgG for CD81, and 1:2000 HRP conjugated anti-rabbit IgG for calnexin). Blots were developed with 1:1 solution of Radiance Chemiluminescent Substrate and Luminol/Enhancer (Azure Biosystems, Dublin, CA, USA). A c300 imaging system (Azure Biosystems) was used to visualize the blots, and images acquired were quantified using the ImageJ software version 1.52a.

### 2.4. Nanoparticle Tracking Analysis (NTA)

EV size distribution curves and concentration measurements were carried out by NTA using a Nanosight NS300TM (Malvern Instruments, Malvern, UK). For NTA analysis, BDEV pellets were resuspended in 100 μL PBS of which 10 μL of the sample was diluted to 1:100–1:1000 in PBS prior to measurements. All samples were loaded with the laser module outside the instrument. As the sample was loaded, care was taken to avoid air pockets. The machine was equipped with a 488 nm laser and a syringe pump system, with a pump infusion speed of 20. The standard measurement option was selected for the scripted workflow to capture videos. The number and duration of captures were set to 5 and 60 s, respectively. The base filename and location were selected prior to starting the run. The camera level was set at 11. Background measurements were performed with filtered PBS, which revealed the absence of any kind of particles. Five video recordings were carried out for each EV preparation with a duration of the 60 s with frame rates of 25 frames/s. Once the videos were recorded, the NTA 3.1 software version was used to analyze the sample videos. For analysis, the screen gain was set to 1.0, and the detection threshold was adjusted to set the minimum brightness of pixels to be considered. At the end of the analyses, the dilution factor for the samples was updated before data export.

### 2.5. Transmission Electron Microscopy (TEM)

TEM analysis was performed as described in our recently published study [31]. Briefly, BDEVs pellets were resuspended in 1× PBS. Then, 90 µL of TEM fix buffer (2% glutaraldehyde, 2% paraformaldehyde, and 0.1 M phosphate buffer) was mixed with 10 µL of BDEVs suspension. A 10-μL drop of BDEVs-buffer solution was spotted on 200-mesh copper grid coated with formvar and silicon monoxide and allowed to incubate for 2 min. The excess solution was taken off by filter paper, and the thin film of sample was allowed to air dry for 2 min. Using a NanoVan (Nanoprobes, New York, NY, USA), the samples were negatively stained on the grid. The excess negative stain was then taken off by filter paper and allowed to dry for at least 1 min before being imaged. Grids were assessed on a Tecnai G2 TEM (built by FEI, Hillsboro, OR, USA) functioned at 80 kV. An advanced microscopy techniques digital imaging system was used to acquire digital images.

### 2.6. Total RNA Isolation, cDNA Preparation, and Quantitative Real-Time PCR

Total RNA was isolated from individual rats from both sexes and treatment groups using the Direct-Zol RNA kit (Zymo Research, Irvine, CA, USA) based on the manufacturer’s protocol. Quantitation of extracted total RNA and RNA integrity was determined by Epoch (BioTek, Winooksi, VT, USA). cDNA was prepared from the respective samples using the Superscript IV kit (Invitrogen, Waltham, MA, USA), and thereafter RT-PCR was performed using TaqMan Custom Array Plates for genes involved with the endosomal sorting complexes required for transport (ESCRT) pathways. The expression of the following 28 genes were analyzed: HGS, STAM, STAM2, MVB12A, MVB12B, TSG101, UBAP, VPS28, VPS37A, VPS37B, VPS37C, VPS37D, SNF8, VPS25, VPS36, CHMP2A, CHMP2B, CHMP3, CHMP4B, CHMP4C, CHMP6, VPS4A, VPS4B, Cers2, Cers3, Cers4, Cers5, and Cers6 with glyceraldehyde 3-phosphate dehydrogenase (GAPDH) as control. The Delta-delta Ct method was used to calculate fold change and statistical significance [30]. 

### 2.7. Mass Spectrometry/Proteomics

Label free quantitative mass spectrometry was performed as described in an earlier study [33]. A 50 µg sample of protein per group was taken, and detergent was removed by chloroform/methanol extraction, and the protein pellet was re-suspended in 100 mM ammonium bicarbonate and digested with MS-grade trypsin (Pierce, Dallas, TX, USA) overnight at 37 °C. Peptides were cleaned with PepClean C18 spin columns (ThermoFisher, Waltham, MA, USA) and re-suspended in 2% acetonitrile (ACN) and 0.1% formic acid (FA). Of each sample, 500 ng was loaded onto trap column Acclaim PepMap 100 75 µm × 2 cm C18 LC Columns (Thermo Scientific, Waltham, MA, USA) at flow rate of 4 µL/min, then separated with a Thermo RSLC Ultimate 3000 (Thermo Scientific) on a Thermo Easy-Spray PepMap RSLC C18 75 µm × 50 cm C-18 2 µm column (Thermo Scientific) with a step gradient of 4–25% solvent B (0.1% FA in 80% ACN) from 10–130 min and 25–45% solvent B for 130–145 min at 300 nL/min and 50 °C with a 180 min total run time. Eluted peptides were analyzed by a Thermo Orbitrap Fusion Lumos Tribrid (Thermo Scientific) mass spectrometer in a data-dependent acquisition mode. A full survey scan MS (from m/z 350–1800) was acquired in the Orbitrap with a resolution of 120,000. The AGC target for MS1 was set as 4 × 105 and ion filling time set as 100 ms. The most intense ions with charge state 2–6 were isolated in a 3 s cycle and fragmented using HCD fragmentation with 40% normalized collision energy and detected at a mass resolution of 30,000 at 200 m/z. The AGC target for MS/MS was set as 5 × 104 and ion filling time set 60 ms dynamic exclusion was set for 30 s with a 10-ppm mass window. Protein identification was performed by searching MS/MS data against the swiss-port *Rattus norvegicus* protein database, using the in-house mascot 2.6.2 (Matrix Science, Boston, MA, USA) search engine. The search was set up for full tryptic peptides with a maximum of two missed cleavage sites. Acetylation of protein N-terminus and oxidized methionine were included as variable modifications, and carbamidomethylation of cysteine was set as a fixed modification. The precursor mass tolerance threshold was set 10 ppm for, and maximum fragment mass error was 0.02 Da. The significance threshold of the ion score was calculated based on a false discovery rate of ≤1%. Qualitative analysis was performed using Progenesis QI proteomics 4.1 (Nonlinear Dynamics, Milford, MA, USA).

### 2.8. Bioinformatics Analysis

Gene Ontology (GO) analysis of differentially expressed proteins was performed using the Cytoscape plugin ClueGO [34]. Only proteins exhibiting significantly different expression across nicotine vs. saline comparison were included in this analysis. Biological process, molecular function, along with KEGG pathways, were included for GO enrichment analysis. Heatmaps were generated for differentially expressed proteins in each comparison using the function heatmap.2 in the R (version 3.6.0) package gplots.

### 2.9. Statistical Analysis

For proteomics analysis, after normalization, a student t-test was done to identify proteins showing significant differences between groups (saline versus nicotine for each of the sexes). Proteins that had at least two unique peptides and a t-test *p*-value < 0.05 were considered significant. For the ESCRT analysis, a 2-way ANOVA followed by Sidak’s multiple comparison test with *p* < 0.05 was considered significant. All statistical tests were performed and analyzed with GraphPad Prism (La Jolla, CA, USA); data represented as mean  ±  SEM on the graphs.

## 3. Results

### 3.1. Nicotine Exposure Increases BDEV Size Which Is More Pronounced in Females

Our previously published study revealed that male and female rats readily self-administered nicotine, while yoked saline maintained very little responding [29]. In addition, we also reported that female rats had a significantly higher total nicotine intake than males, which is consistent with previously published papers [11,14,15,35,36,37]. Based on this observation, we hypothesized that nicotine intake impacts the BDEV sizes in males and females. Accordingly, we isolated BDEV from the different treatment groups using a sucrose density gradient ultracentrifugation method, as described in our previous studies [30,31,32]. To further ascertain the purity of isolated BDEV, we performed Western blot analysis using positive and negative EV markers, which were established by the International Society for Extracellular Vesicles [38]. Figure 2A,B shows the presence of the positive markers Hsp70, Flotillin-1, and CD81 in the BDEVs and absence of negative marker calnexin. NTA analysis showed a slight increase in the number of BDEVs released in both males and females post nicotine exposure but were overall not significant (Figure 2C).

TEM analysis of BDEV revealed a cup-like appearance of vesicles. While the BDEV sizes for the saline animals were around 85 nm and 140 nm, nicotine self-administration caused an increase in vesicle size, which was more pronounced in the females (Figure 3A,B).

### 3.2. Nicotine Increases BDEV Biogenesis Which Are More Pronounced in Females

Based on the increase in the BDEV size with nicotine, we then assessed whether EV biogenesis was also altered between males and females. EV biogenesis depends on several genes, including the ESCRT pathway [39,40] and ESCRT independent pathways such as the ceramide synthesis pathway [41]. Total RNA from males and females from the saline and nicotine groups was analyzed using a multiplex RT-PCR array. Several genes involved in the ESCRT dependent and independent pathways were significantly upregulated and more pronounced in the females. Specifically, genes belonging to the families ESCRT-0: HGS, STAM1 and 2; ESCRT-I: MVB12A and B, TSG101, UBAP1, VPS37A and D; ESCRT-II: VPS25; ESCRT-III: CHMP2A and 6; and the disassembly complex: VPS4A and B were significantly elevated in the female rats post nicotine self-administration. Further, we also found an upregulation of genes in the ceramide synthesis pathway such as CERS2, CERS4, 5, and 6 in the females (Figure 4). Together, these data indicate that the ESCRT and ceramide pathways were affected by chronic nicotine exposure and are further pronounced in females.

### 3.3. Proteomics and Bioinformatics Analysis

To further corroborate that increased BDEV size and altered biogenesis with chronic nicotine intake also leads to changes in BDEV cargo, we used quantitative mass spectrometry-based proteomics to identify changes in the BDEV proteins between males and females. A total of 2165 and 2051 proteins were identified between males and females, respectively (Appendix A). Further employing a criterion of 2+ unique peptides, and *p* < 0.05, we identified 85 proteins and 31 proteins to be differentially expressed between males and females respectively (Appendix A). Figure 5A shows the Venn diagram of the differentially expressed proteins with saline and nicotine treatment among males and females. Ten proteins were significantly upregulated with nicotine for females, while six were upregulated in males. On the other hand, 21 proteins were significantly downregulated for females with nicotine, while 79 were downregulated for males. A heatmap was then generated based on the significant differentially expressed proteins between nicotine and saline groups for both male and female as shown in Figure 5B,C. Table 1 shows the distinct significant proteins for each sex to be expressed 1.5-fold up or down with nicotine exposure.

Next, the enriched biological processes of these differentially expressed proteins were determined for both male and female through ClueGO analysis that revealed several enriched biological processes for the males: receptor internalization, synaptic vesicle cycle, glutathione metabolism, GPI anchor metabolic process, GABAergic synapse, cardiac conduction, glial cell proliferation, and N-glycan biosynthesis (Figure 6). For the females, the enriched biological processes included bile secretion, gastric secretion, and axon regeneration. As shown in Figure 6, there are various gene ontology terms involved at the neuronal level, such as axon regeneration, synaptic vesicle cycle, glial cell proliferation, and GABAergic synapse. Axon regeneration was more specific to the female group, while synaptic vesicle cycle, glial cell proliferation, and GABAergic synapse were specific to the male group. These data suggest that nicotine treatment does have an impact on BDEVs and potentially impairs important neuronal functions.

## 4. Discussion

Our previously published study demonstrated the impact of nicotine on sex differences through a drug self-administration model that mimics human drug-taking behavior [29]. Though male and female rats readily self-administer nicotine, female rats had a significantly higher total nicotine intake than males (+1.35 fold). This result is evident with many previously published works [11,14,15,35,36,37]. While many studies have acknowledged various behavioral factors for observed sex differences associated with chronic nicotine intake, the molecular mechanisms contributing to these sex differences remain poorly explained. To fill this critical knowledge gap, the current study focused on elucidating the role of BDEVs in sex differences associated with nicotine self-administration. EVs are vesicles secreted by cells and carry cellular cargo such as microRNA and proteins. They range from 30 to 120 nm in size and regulate several pathophysiological processes like inflammation and tumor growth [42,43,44], thus denoting their potential as valuable diagnostic markers and therapeutic vehicles [45,46]. 

Studies examining the effect of smoking and EV release have shown an increase in EV number and sizes. For example, one study that assessed the bronchoalveolar lavage from smokers with non-small cell lung cancer showed an increase in EV particle release, which subsequently led to an increase in EV microRNA expression compared to healthy controls [47]. Another study using airway epithelial cells exposed to cigarette smoke extract also showed an increase in EV particle release that was prevented by scavenging thiol anti-oxidants N-acetyl-L-cysteine or glutathione [48]. Corsello et al. also showed airway epithelial cells exposed to environmental tobacco smoke displayed an increase in EV size that further lead to enhanced EV cargo content [49]. Furthermore, a recent study examining the effects of e-cigarettes containing nicotine revealed an increase in endothelial and platelet-derived EVs in healthy volunteers [50]. Our current observation that chronic nicotine treatment caused an increase in both BDEV concentration and size in males and females further bolsters these above-mentioned studies. Interestingly, the differences in BDEV size and concentration were more pronounced in females. One possible reason that could be associated with the increased concentration and sizes of BDEV in females could be due to the higher intake of nicotine. Yet another logical explanation could be the stage of estrous, which has been associated with the observed sex differences with drugs of abuse [51,52]. Data from earlier studies have shown that intact female rats show higher behavioral response to psychostimulants such as cocaine in estrus compared to other stages of the estrous cycle [53] or to males [54]. This observation further illustrates support from another study, which showed females reaching higher breaking points during estrous than during other stages of the estrous cycle on a progressive ratio of cocaine self-administration [55]. 

Another important feature associated with EVs is their biogenesis, which is frequently dysregulated in several pathologies [55]. This biogenesis is mediated by several proteins involved in their maturation and trafficking. One such group of proteins is the endosomal sorting complexes required for transport (ESCRT) machinery as well as members of the Rab family of small GTPases [56]. ESCRT dependent and independent pathways play a major role in EV biogenesis [57]. Four different ESCRTs have been identified, ESCRT 0, I, II, and III [58]. ESCRT 0 recognizes ubiquitinated proteins on the outside of the endosomal membrane [59]. ESCRT I and II are recruited to the cytosolic side of the early endosomes via various stimuli [59]. We identified the ESCRT complex 0 proteins (HGS, STAM1/2), complex I (MVB12A/B, TSG101, UBAP1, VPS37A/D), complex II (VPS25), and complex III (CHMP2A, 6 along with VPS4A/B and VTA1) to be significantly elevated in the females from our qRT-PCR analysis. HRS is responsible for the secretion of small EVs (<50 nm), whereas STAM1/2 are crucial for the secretion of larger sized EVs (>100 nm). The heterogenous population of BDEVs observed in our findings indicates that similar pathways might be responsible for increased EV biogenesis during chronic nicotine treatment and more perturbed in females. ESCRT-I complex mediates the sorting of ubiquitinated cargo protein from the plasma membrane to the endosomal vesicle.

The significant increase in the expression of ESCRT-I proteins, such as MVB12A/B, UBAP1, TSG101, and VPS37A/D, in the females possibly denotes more translocation of the protein cargo to the endosome. In addition, some ESCRT-1 complex proteins such as TSG101 and VPS28 have been linked to autophagy, a process that regulates the removal of dysfunctional cellular components [59]. Although nicotine exposed female rats had an increase in the expression of VPS28, it was not significant. However, the rats with a significant increase in TSG101 expression possibly have increased autophagy. A previous study demonstrated that knockdown of HRS, STAM1, and TSG101 decreased the total exosome secretion [39]. Based on the increased expression of these critical regulators of exosome secretion in the females post nicotine intake, knocking these down could possibly decrease exosome secretion and relevant cargo. It has been suggested that ESCRT I and II are the initiators and drivers of the intraluminal membrane budding, whereas ESCRT III completes this process [58,60,61]. Further, the ESCRT-II-ESCRT-III interaction coordinates the sorting of ubiquitinated cargo with the budding and scission of intralumenal vesicles into multivesicular bodies [61]. The significant up-regulation of the complex II protein VPS25 and of complex III CHMP2A and CHMP6 in the females post nicotine exposure denotes enhanced sorting of ubiquitinated cargo for degradation.

In addition to the ESCRT dependent pathway, independent mechanisms regulating EV biogenesis in the absence of ESCRTs have been documented [62]. One such pathway is ceramide biosynthesis that includes key genes: CERS2, 3, 4, 5, 6, and NSMASE2 have been previously shown to be involved in EV biogenesis and release [62]. Ceramide is one of the lipids critical for exosome formation [63]. Emerging research points to the role of the presence of such lipid enriched molecules on EV membranes in several signaling pathways in disease states such as cancer. Furthermore, a recent study showed increased levels of very long chain C24:1 ceramide in serum EVs of older women with a similar effect in rhesus macaques compared to the respective younger groups suggesting its implication in aging [64]. CERS2 is the primary synthase involved in synthesizing very long chain C24:1 ceramide through the de novo pathway, whereas NSMASE2 is primarily involved in the production of ceramide by hydrolysis of sphingomyelin. Elevated ceramide levels have been associated with poor cardiovascular health and memory impairment in older adults [65,66]. In our current study, the expression levels of CERS2 along with 4, 5, and 6 were significantly upregulated in the females after nicotine self-administration, while CERS3 was downregulated, albeit not significantly, which could suggest the vulnerability of females dependent on nicotine to faster aging.

EVs contain selective repertoires of proteins, RNAs, lipids, and metabolites that moderate signaling pathways in the recipient cells [67,68]. Our current study did identify differentially expressed proteins between males and females post nicotine exposure. Notably, the number of significantly altered proteins after nicotine self-administration was lower in females compared to males. A logical explanation could be the larger BDEV sizes, as observed with TEM analysis. In addition, the altered biogenesis, as observed from the elevated expression of key ESCRT dependent and independent proteins, could possibly lead to enhanced release of protein cargo and targeted to the endolysosome for degradation.

Among the females, Aquaporin-1 (AQP1), a membrane channel important for maintaining the osmotic gradient, was upregulated 2-fold with nicotine exposure. A recent study demonstrated that nicotine induces lower urinary tract symptoms that reduce bladder blood flow and subsequently inducing urothelial hypoxia [69]. From a CNS perspective, AQP1 increase on BDEV by nicotine could possibly be associated with inducing alterations in cerebral vasculature that further could lead to brain dysfunction.

Another protein that was downregulated (–1.5 fold) with nicotine in females is the proteasome subunit beta (PSMB) essential for maintaining protein homeostasis. Previous research has shown that nicotine, at the concentration of smokers, significantly reduces proteasomal activity [12]. Since proteasome catalyzes the degradation of many proteins associated with various biological processes, downregulation of PSMB possibly results in the accumulation of more toxic cargo protein on the BDEVs, which subsequently could exacerbate neurodegeneration in the females.

Amongst the males, Cyclin Y (CCNY) was upregulated 2.1 fold. CCNY plays a significant role in regulating the cell cycle and transcription. Previously published research has established that cyclins are associated with nicotine addiction [70]. CCNY has also been shown to impact several biological processes, including learning and memory [71]. This suggests that males are potentially more vulnerable to higher cognitive decline with nicotine dependency. Furthermore, among the males, we found the expression of mesoderm-specific transcript homolog protein (MEST) to be downregulated 2.7-fold. MEST consists of hydrolase function and may play a role in development. Currently, there is no literature examining the impact of nicotine addiction and MEST, but further studies examining the effect of nicotine on its hydrolase activity may possibly yield more mechanistic insights.

Next, we examined the key biological processes associated with differentially expressed proteins. One key biological process we observed in the males was receptor internalization. Many studies have shown that chronic exposure to nicotine impacts the number and localization of nAChRs. Previous research has shown that chronic exposure to nicotine results in an increase in nicotine binding sites [72,73,74]. There are various theories regarding the mechanism for the upregulation of nAChR expression, which includes a decrease in receptor internalization. The decrease in receptor internalization is due to conformational changes caused by nicotine, which prevents the removal of the receptors from cell surfaces [70,75]. The enrichment of this particular biological process could mean that male rats, which also self-administer nicotine may have increased binding and thus decreased removal of the nAChRs from the cell surfaces. Another enriched biological process among male rats is the synaptic vesicle cycle, which is associated with neurotransmission. Earlier research has shown that numerous addictive drugs, including nicotine, enhance neuronal firing beyond the norm and alter the release probability from the presynaptic terminal [71]. While smoking, the nicotine levels are assumed to desensitize the dopamine terminal presynaptic nAChR, but this is overcompensated for by the high-frequency neuronal firing activity, and as a result, nicotine promotes dopamine neurotransmission. With the enhancement of dopamine neurotransmission, there is also an increase of the readily releasable pool of synaptic vesicles [71], thus suggesting enhanced synaptic vesicle dynamics in male rats after nicotine exposure.

A third enriched biological process among male rats exposed to nicotine is glutathione metabolism. Glutathione plays an important role in antioxidant defense, nutrient metabolism, and regulation of cellular events. A deficiency of glutathione results in oxidative stress which leads to the pathogenesis of many diseases like Alzheimer’s and Parkinson’s. Research has shown that cigarette smoking leads to an accumulation of reactive oxygen species (ROS), which potentially leads to osteoporosis due to a decrease in bone mineral density and impaired fracture healing [76]. Another study has shown that cigarette smoke extract affects the ROS level as well as decreases the reduced glutathione (GSH) concentration along with altering expression of several antioxidant enzymes in human bronchial epithelial cells [77]. Many animal models have shown the same effect of a decrease in reduced glutathione. One study found that the levels of GSH were reduced in the cerebellum, medulla oblongata, and hemisphere regions after young Wistar rats received a single dose of nicotine [78]. Another study with mice showed a significant decrease in GSH in the lungs after daily nicotine injections for several weeks [79]. Based on these studies, BDEVs could consist of key proteins associated with glutathione metabolism.

While the male group revealed many enriched biological processes, there were only three revealed among the female group: bile secretion, axon regeneration, and gastric acid secretion. There are numerous studies that have examined the effect of nicotine on gastric acid secretion, however, the role of nicotine remains elusive. Studies have shown that chronic nicotine administration increases the gastric acid output, which is consistent with the fact that nicotine administration can lead to an increase in muscarinic receptor sensitivity, which as a result, leads to gastric acid secretion [80]. Research has also shown the amount of gastric acid secretion is associated with the number of cigarettes smoked, and smoking over a period a time results in pentagastrin-induced gastric acid output [81]. On the other hand, researchers have also shown that gastric acid secretion is inhibited by nicotine [80]. One study has shown that the gastric acid output after one hour of vagal stimulation was lower among human subjects on smoking days compared to non-smoking days [80].

Gastric acid secretion goes hand in hand with the second enriched biological process among females: bile secretion. Smoking has been shown to increase bile salt reflux rate and bile salt concentration, which, as a result, increases the risk for duodenogastric reflux leading to ulceration [82]. Research with the addictive component itself, nicotine, has also shown an increase in bile acid concentration. One study with rabbits receiving low, medium, and high doses of nicotine for 14 days revealed an increase in bile acid concentration across all groups with the highest being of the low dose nicotine group [83]. Our data shows that smoking leads to an increase in gastric acid and bile acid secretion, predisposing females dependent on nicotine to cancers and other diseases [84]. The final enriched biological process among females is axon regeneration. If neuronal damage occurs, nicotine is shown to have more negative effects in the healing process, including axon regeneration [85]. However, research has also shown nicotine to be beneficial in relation to some diseases like multiple sclerosis. Multiple sclerosis is a disease that leads to CNS inflammation, demyelination, and axonal damage. Nicotine exposure in mice has shown that nicotine “betters” the axonal damage meaning it helps in the regeneration of axons [86]. In the current context, enrichment of this pathway in the BDEV protein cargo of females possibly indicates a neuroprotective effect against the increased intake of nicotine.

To summarize, this current study has identified unique enriched biological processes amongst BDEVs of male and female rats post nicotine self-administration. Future works include the validation of unique differentially expressed proteins found within these enriched biological processes for each sex. Additionally, studies aimed at dissecting mechanisms associated with the various enriched biological processes will lead to significant insight into understanding the sex differences associated with nicotine self-administration.

## 5. Conclusions

To summarize, this current study for the first time showed alterations in BDEV biogenesis and protein cargo by self-administration of nicotine in both male and female rats, which were further pronounced in the latter. Furthermore, we identified sex-specific BDEV protein signatures, including key biological processes that could further help our understanding of the mechanisms of sex differences with nicotine dependency.

## Figures and Tables

**Figure 1 cells-09-01883-f001:**
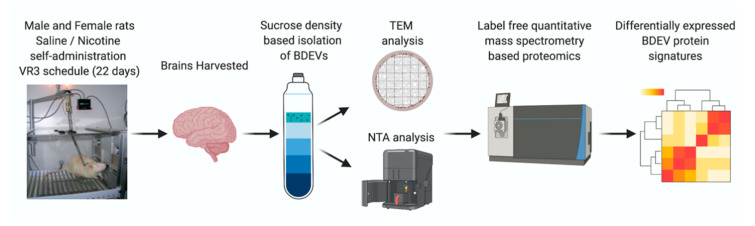
Scheme of experimental in vivo design. A visual representation of the experimental process beginning with nicotine self-administration followed by brain isolation, EV extraction, and proteomics analysis.

**Figure 2 cells-09-01883-f002:**
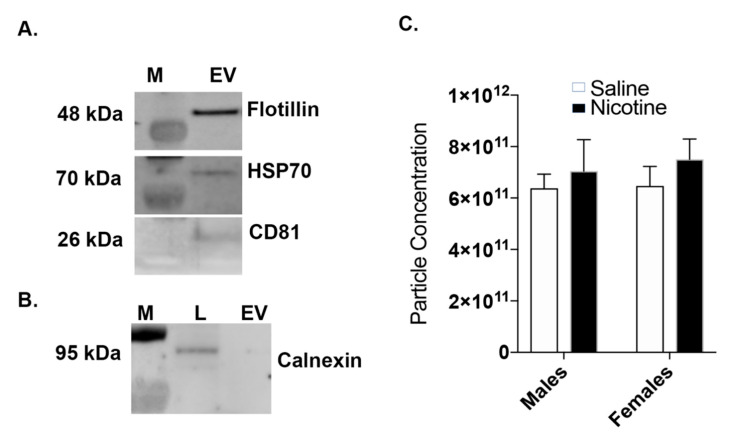
Validation of BDEV purity. (**A**) Western blot analysis on BDEV isolated from a control saline animal shows the expression of the positive markers CD81, flotillin-1, and Hsp70. (**B**) The negative marker calnexin was enriched in the whole tissue lysate (L) but absent in the EVs fraction (EV). (**C**) Nano tracking analysis (NTA) shows an increase in BDEV particle concentration in both and male and female rats exposed to nicotine. However, no significant changes in concentrations of isolated BDEVs from the two treatment groups between the sexes was observed as determined by a 2-way ANOVA followed by Sidak’s multiple comparison test.

**Figure 3 cells-09-01883-f003:**
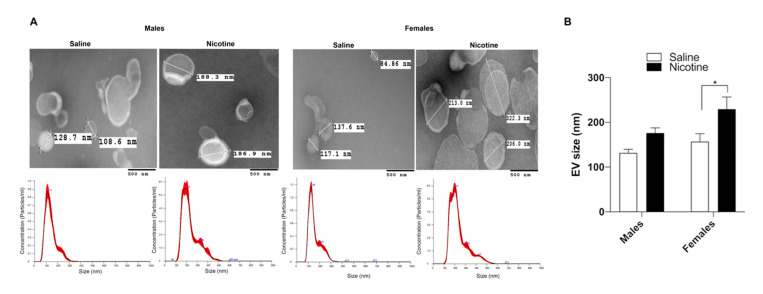
Characterization of BDEV. (**A**) BDEV isolated using a sucrose density gradient from the three groups were characterized using TEM that revealed increased sizes in both males and females which was larger in the latter post nicotine treatment. NTA analysis showing the different size distribution. (**B**) Average BDEV sizes. Average BDEV sizes for each group is represented as mean ± SEM, * *p* < 0.05 as determined by a 2-way ANOVA followed by Sidak’s multiple comparison test.

**Figure 4 cells-09-01883-f004:**
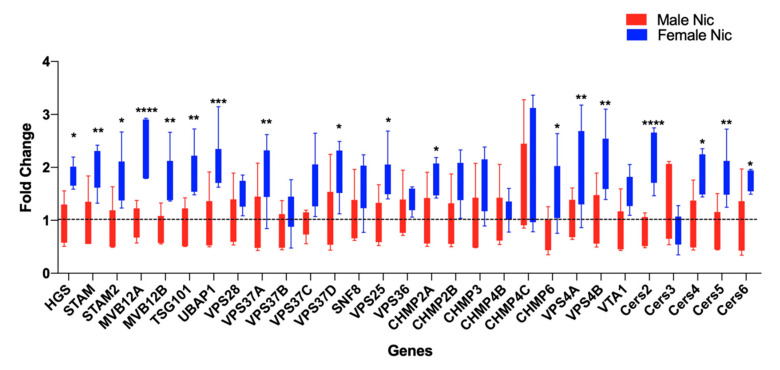
Chronic nicotine treatment increases the biogenesis and the expression of ESCRT dependent and independent pathway proteins. A custom qRT-PCR panel for EV biogenesis genes revealed several genes involved in the ESCRT dependent (complexes 0, I, II, and III) and independent pathways to be significantly upregulated in the females post nicotine self-administration. Data represented as mean ± SEM, n = 4–6 animals per group, **** *p* < 0.0001, *** *p* < 0.001, ** *p* < 0.01, * *p* < 0.05 (adjusted *p* value) as determined by a multiple t-test followed by Holm–Sidak correction.

**Figure 5 cells-09-01883-f005:**
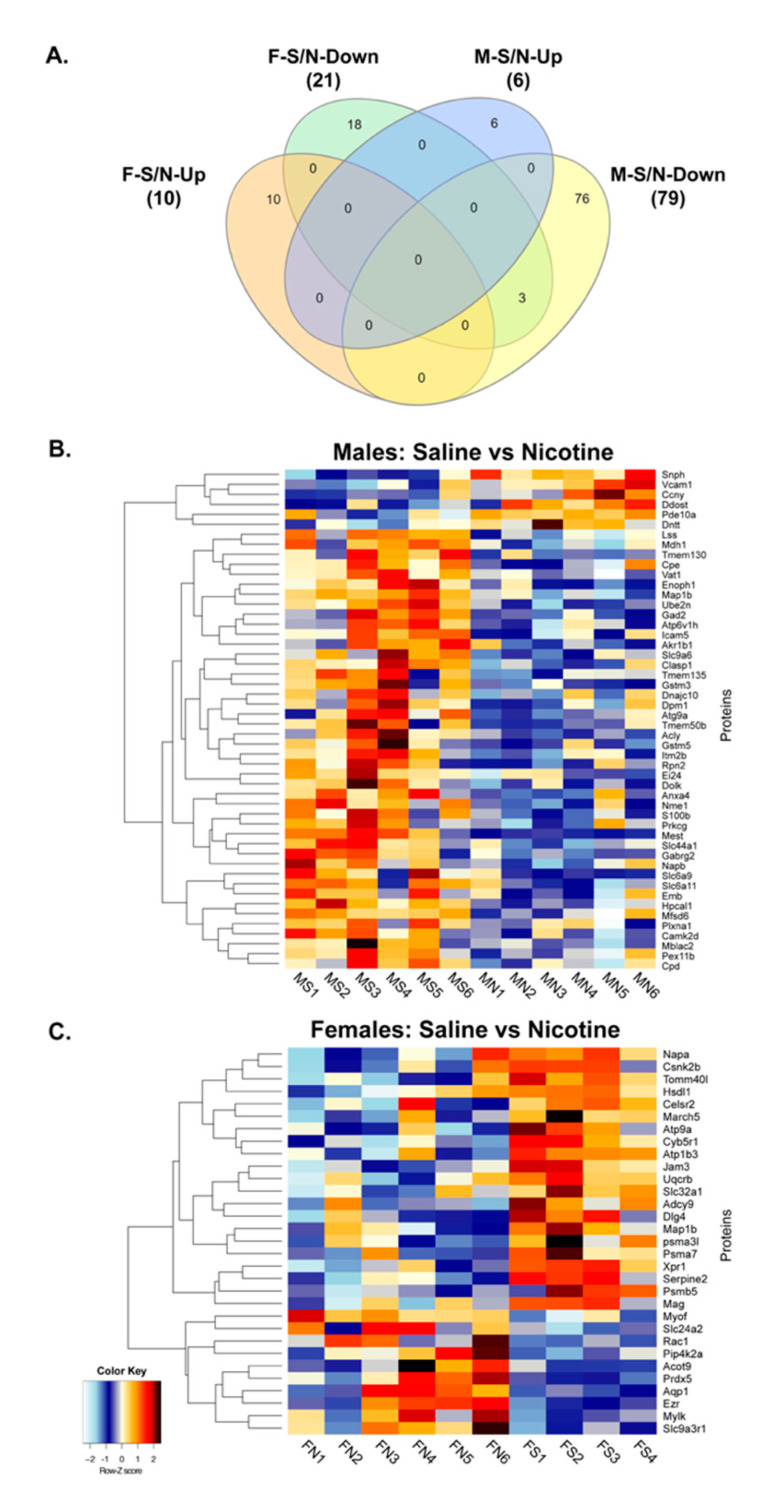
Differential expression of BDEV proteins in male and female rats post nicotine self-administration. (**A**) Venn diagram showing the differentially expressed BDEV proteins identified from the proteomics screen. (**B,C**) Heatmap showing the top differentially expressed proteins between the two sexes post nicotine self-administration compared to saline. MS, MN; FS, FN—males, females self-administering saline and nicotine respectively.

**Figure 6 cells-09-01883-f006:**
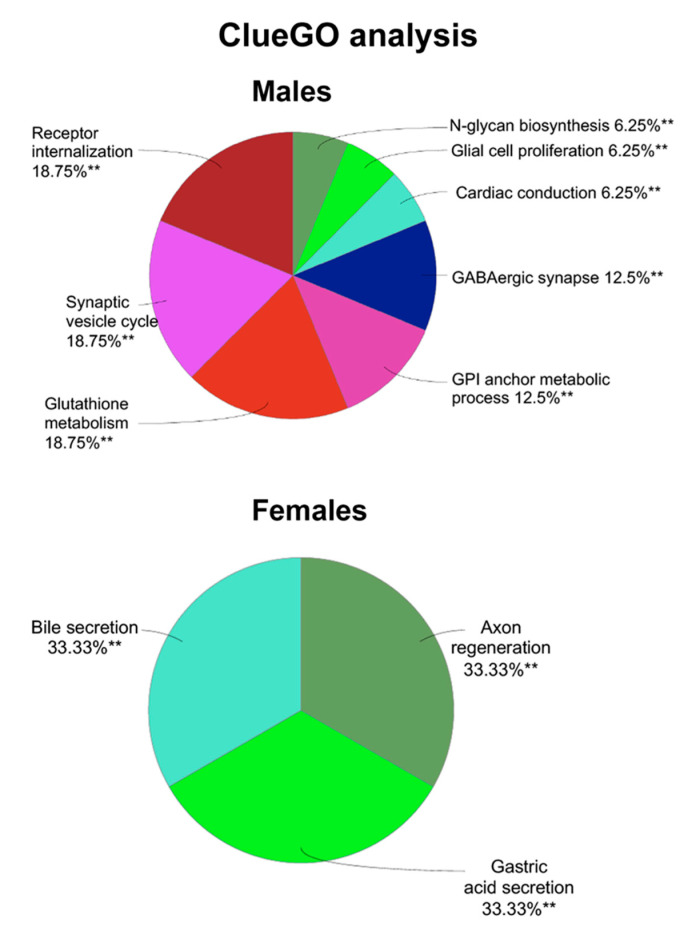
Mapping of biological processes. ClueGO analysis showing enriched biological processes in males and females after nicotine self-administration. The asterisks represent the group term *p*-value representing each category. ** *p* < 0.01.

**Table 1 cells-09-01883-t001:** Significant proteins differentially expressed between males and females after nicotine self-administration. A criterion of 2+ unique peptides and fold change of >1.5 was used to select the potential hits.

Group	Accession Number	Gene Name	Fold Change
**Males**	F1MA89	Cyclin Y	+2.1
	A0A0G2K127	Vascular cell adhesion protein 1	+1.6
	P62749	Hippocalcin-like protein 1	−1.5
	F8WFM2	Beta-soluble NSF attachment protein	−1.5
	Q5XIE8	Integral membrane protein 2B	−1.5
	Q9EQX9	Ubiquitin-conjugating enzyme E2	−1.5
	Q9Z1B2	Glutathione S-transferase Mu 5	−1.5
	D3Z981	Plexin A1	−1.5
	Q05683	Glutamate decarboxylase 2	−1.6
	Q05982	Nucleoside diphosphate kinase A	−1.6
	P04631	Protein S100-B	−1.6
	P15205	Microtubule-associated protein 1B	−1.6
	P31647	Sodium- and chloride-dependent GABA transporter 3	−1.6
	A0A0G2K9J2	V-type proton ATPase subunit H	−1.6
	P08009	Glutathione S-transferase Yb-3	−1.6
	Q6PW52	GABA-A gamma2 long isoform	−1.8
	A0A0G2K5E7	ATP-citrate synthase	−1.9
	D4A8N1	Dolichol-phosphate mannosyltransferase subunit 1	−2.0
	M0R830	Mesoderm-specific transcript homolog protein	−2.7
**Females**	P29975	Aquaporin-1	+2.0
	A0A0G2K890	Ezrin	+1.8
	Q9R0I8	Phosphatidylinositol 5-phosphate 4-kinase type-2 alpha	+1.7
	Q5U2 × 8	Acyl-CoA thioesterase 9	+1.6
	G3V7Q6	Proteasome subunit beta	−1.5

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
