# Peer review of "Role of Brain Derived Extracellular Vesicles in Decoding Sex Differences Associated with Nicotine Self-Administration"

_cells, 2020, doi:10.3390/cells9081883_

Round 1

Reviewer 1 Report

In this paper, the authors focus on determining the molecular mechanisms involved in the differential effects of nicotine between males and females rats in EV secretion.

Major comments

The methodology applied to isolate extracellular vesicles (EV) should be complemented with other techniques, as CSF or plasma measurement, because it is possible that the fractions obtained by ultracentrifugation contain a mixture of intracellular and extracellular vesicles. With this experimental approach is not possible evaluate secretion or release of vesicles.

Minor comments

In figure 2. I suggest includes a graph to show the differences of vesicles size between the experimental groups and if these differences are significant statistically.

Authors should highlight that the results obtained by proteomic analysis are associated with biogenesis of EVs.

These alterations can be associated to transcriptional changes trigger by nicotine administration. Furthermore, the authors should speculate the possible effect of nicotine on transcription factors that could be involve in the changes observed and that explain the differential effects in females compared to males.

The authors must be include the validation of target genes/proteins by qPCR or Western blot.

Author Response

Major comments

The methodology applied to isolate extracellular vesicles (EV) should be complemented with other techniques, as CSF or plasma measurement, because it is possible that the fractions obtained by ultracentrifugation contain a mixture of intracellular and extracellular vesicles. With this experimental approach is not possible evaluate secretion or release of vesicles.

Great point. The goal here was to identify brain derived EVs as potential markers associated with nicotine dependency between male and female rats. The isolation technique using sucrose density is well established in our lab as evidenced by our previous publications. We however understand that the methodology to isolate extracellular vesicles should be complemented with other technique, but there are various published studies on those fronts.

Minor comments

In figure 2. I suggest includes a graph to show the differences of vesicles size between the experimental groups and if these differences are significant statistically.

Thank you for bringing this to our attention. A bar graph showing the differences of vesicle sizes between experimental groups including the relevant statistics has now been added as Figure 3B.

Authors should highlight that the results obtained by proteomic analysis are associated with biogenesis of EVs.

We have added the necessary information on these lines including discussing the new data (Figure 3B), showing a significant increase in the EV size in females. This observation dovetails with the significant changes in key genes associated with EV biogenesis in the females post nicotine exposure.

These alterations can be associated to transcriptional changes trigger by nicotine administration. Furthermore, the authors should speculate the possible effect of nicotine on transcription factors that could be involve in the changes observed and that explain the differential effects in females compared to males.

Terrific comment. However, given our emphasis on BDEV, we believe this excellent suggestion could well form a strong premise for future studies.

The authors must be include the validation of target genes/proteins by qPCR or Western blot.

Great point, the tissues sample used for this study were obtained from a previously published study and due to the non-availability of tissue (see point 2 under R2) we are unable to validate the target genes/proteins by qPCR or Western blot. We hope the reviewer understands the limitation associated with animal tissues involving such long duration of treatments.

Reviewer 2 Report

Experimental animal models have shown nicotine associated sex differences. It is much less clear whether extracellular vesicles (EV) are regulators in this effect. In the manuscript "Role of brain derived extracellular vesicles in decoding sex differences associated with nicotine self-administration" the authors used a multiplex RT-PCR array and Mass Spectrometry to assess differences in sex-specific changes in BDEV biogenesis, protein cargo signatures, and molecular pathways associated with long-term nicotine self-administration.
The present work is based on an interesting issue, the study was properly designed and it technically sounds. However, there are some concerns that should be better addressed in order to improve the quality of the manuscript.

Comments and Suggestions for Authors

1) methods. It will be useful to make a scheme that explains the experimental design in vivo.
2) The nature of n's should be clarified. For some experiments, the n's are sufficiently low, ie 4-6, that risk of a Type I error is increased.
3) The manuscript is quite long, some of this is necessary. However, the discussion includes a number of topics and speculations that might be shortened.

Author Response

Reviewer 2

1) Methods. It will be useful to make a scheme that explains the experimental design in vivo.

Excellent point made. A scheme that explains the experimental design in vivo has now been included as Figure 1 in the revised manuscript.

2) The nature of n's should be clarified. For some experiments, the n's are sufficiently low, ie 4-6, that risk of a Type I error is increased.

The n’s per experimental group was n= 4-6 and have been clarified in the manuscript. As for the other comment on the low n-size, we would like to reiterate that due to the non-availability of tissue, (since these study was from a previously published work from our group see line 111), we had to perform the experiments based on what was available. We hope the reviewer understands the limitation associated with animal tissues involving such long duration of treatments.

3) The manuscript is quite long, some of this is necessary. However, the discussion includes a number of topics and speculations that might be shortened.

We kindly thank you for your suggestion. The discussion section of this manuscript is detailed to better address the potential reasons as to the impact of nicotine self-administration on brain-derived extracellular vesicles in male’s vs females. We hope this revision has now quelled any concerns that the reviewer believes are a speculation. 

Reviewer 3 Report

The authors investigated the role of brain-derived extracellular vesicles (BDEV) in decoding sex differences associated with nicotine self-administration in an animal experimental model. The search for factors that may influence sex differences in addiction to nicotine, is a field of increasing research interest and of great translational potential to clinical practice.

The study presents sound methodology and the manuscript is well written. Nevertheless, despite the importance of this study, the manuscript presents points requiring clarification.

Comments:

#1 – On page 2, there are some very short paragraphs that should be rearranged.

# 2 – Introduction should summarize epigenetic factors that have hitherto been investigated as modulators of molecular pathways related to sex differences concerning addiction to nicotine.

# 3 – The authors could have determined circulating sex-hormones levels to evaluate potential association with differences found in size and biogenesis of EVs among males and females. Also, gonadal hormone determination could establish at what point of the reproductive cycle the females were when they were sacrificed and brain tissue frozen. The phase of the female reproductive cycle may affect corporal water content and, hence, this may have influenced BDEV biogenesis, size and content. Indeed, it has been reported that sex hormones, particularly progesterone, may be related to brain edema. Of notice, the authors found that aquaporin-1 (a plasma membrane protein involved in water permeation across biological membranes) presented +2-fold expression in females. This topic should be discussed.

# 4 – Since the frozen brain tissue used in the present study was obtained from a previous study (conducted in 2018 according to the reference), the authors should detail the methodological conditions used, then, for brain tissue collection and freezing (i.e. could the methodological conditions previously applied for tissue preservation have influenced RNA degradation?).

#5 – The discussion is too long and speculative, yet some important points related to the findings of the study were not discussed.

#6 – It would be interesting, as this is a study with translational potential, that the authors discussed if a rat-model of self-administration of nicotine may represent the complexity of epigenetic factors involved in human sex differences concerning addictive behavior in different cultures worldwide.

Author Response

Reviewer 3

#1 – On page 2, there are some very short paragraphs that should be rearranged.

Thank you for this suggestion. The short paragraphs have been rearranged.

# 2 – Introduction should summarize epigenetic factors that have hitherto been investigated as modulators of molecular pathways related to sex differences concerning addiction to nicotine.

AND

#6 – It would be interesting, as this is a study with translational potential, that the authors discussed if a rat-model of self-administration of nicotine may represent the complexity of epigenetic factors involved in human sex differences concerning addictive behavior in different cultures worldwide.

We are really appreciative of the reviewer for bringing out this excellent suggestion. Indeed, examining the role of epigenetic factors in relation to sex differences concerning nicotine addiction are important and this indeed is a premise we plan to further explore as potential future studies.

# 3 – The authors could have determined circulating sex-hormones levels to evaluate potential association with differences found in size and biogenesis of EVs among males and females. Also, gonadal hormone determination could establish at what point of the reproductive cycle the females were when they were sacrificed and brain tissue frozen. The phase of the female reproductive cycle may affect corporal water content and, hence, this may have influenced BDEV biogenesis, size and content. Indeed, it has been reported that sex hormones, particularly progesterone, may be related to brain edema. Of notice, the authors found that aquaporin-1 (a plasma membrane protein involved in water permeation across biological membranes) presented +2-fold expression in females. This topic should be discussed.

Excellent point. Our introduction referenced literature that addresses the impact on circulating gonadal hormones on nicotine addiction. Determining circulating sex hormones levels may be examined in further studies. Also, the potential impact/role of aquaporin-1 is now presented in the discussion section (see lines 566-571) in the revised manuscript.

# 4 – Since the frozen brain tissue used in the present study was obtained from a previous study (conducted in 2018 according to the reference), the authors should detail the methodological conditions used, then, for brain tissue collection and freezing (i.e. could the methodological conditions previously applied for tissue preservation have influenced RNA degradation?).

Upon harvesting of the brains, the tissue was immediately frozen in liquid nitrogen and stored at -80°C.

#5 – The discussion is too long and speculative, yet some important points related to the findings of the study were not discussed.

We have revised the discussion section based on this comment including adding emphasis on the four proteins (two each from the two sexes) from Table 1: Cyclin Y, Mesoderm-specific transcript homolog protein, Aquaporin-1, Proteasome subunit beta.

Round 2

Reviewer 1 Report

The authors respond successful my questions. However, this study is limited in its contribution of understand the molecular mechanism involve in the sex differencial nicotine effect due to the validation abscent of gene or pathways to explain this phenomenon.

Author Response

Minor comment

The authors respond successful my questions. However, this study is limited in its contribution of understand the molecular mechanism involve in the sex differential nicotine effect due to the validation absent of gene or pathways to explain this phenomenon.

We understand this concern of the reviewer but, as previously stated in our earlier rebuttal, due to the non-availability of the tissues we unfortunately could not validate the target genes/proteins at this point. We believe the reviewer can understand this limitation associated with animal tissues including the long duration of treatments which are proposed in this study.